# Infection-exposure in infancy is associated with reduced allergy-related disease in later childhood in a Ugandan cohort

Lawrence Lubyayi[1,2]*, Harriet Mpairwe[1,3], Gyaviira Nkurunungi[1,4], Swaib A Lule[5], Angela Nalwoga[1], Emily L Webb[6], Jonathan Levin[2], Alison M Elliott[7]

[1]Immuno-modulation and Vaccines Programme, Medical Research Council/Uganda Virus Research Institute and London School of Hygiene and Tropical Medicine Uganda Research Unit, Entebbe, Uganda; [2]Division of Epidemiology and Biostatistics, School of Public Health, University of the Witwatersrand, Johannesburg, South Africa, Johannesburg, South Africa; [3]Department of Non-Communicable Disease Epidemiology, London School of Hygiene and Tropical Medicine, London, United Kingdom; [4]Department of Infection Biology, London School of Hygiene and Tropical Medicine, London, United Kingdom; [5]Institute for Global Health, University College London, London, United Kingdom; [6]MRC International Statistics and Epidemiology Group, Department of Infectious Disease Epidemiology, London School of Hygiene and Tropical Medicine, London, United Kingdom; [7]Department of Clinical Research, London School of Hygiene and Tropical Medicine, London, United Kingdom

## Abstract

**Background:** Lack of early infection-exposure has been associated with increased allergy-related disease (ARD) susceptibility. In tropical Africa, little is known about which infections contribute to development of ARDs, and at which time.

**Methods:** We used latent class analysis to characterise the early infection-exposure of participants in a Ugandan birth cohort and assessed ARDs in later childhood.

**Results:** Of 2345 live births, 2115 children (90%) had data on infections within the first year of life while 1179 (50%) had outcome data at 9 years. We identified two latent classes of children based on first-year infection-exposure. Class 1 (32 % membership), characterised by higher probabilities for malaria (80%), diarrhoea (76%), and lower respiratory tract infections (LRTI) (22%), was associated with lower prevalence of wheeze, eczema, rhinitis, and *Dermatophagoides* skin prick test (SPT) positivity at 9 years. Based on 5-year cumulative infection experience, class 1 (31 % membership), characterised by higher probabilities for helminths (92%), malaria (79%), and LRTI (45%), was associated with lower probabilities of SPT positivity at 9 years.

**Conclusions:** In this Ugandan birth cohort, early childhood infection-exposure, notably to malaria, helminths, LRTI, and diarrhoea, is associated with lower prevalence of atopy and ARDs in later childhood.

**Funding:** This work was supported by several funding sources. The Entebbe Mother and Baby Study (EMaBS) was supported by the Wellcome Trust, UK, senior fellowships for AME (grant numbers 064693, 079110, 95778) with additional support from the UK Medical Research Council. LL is supported by a PhD fellowship through the DELTAS Africa Initiative SSACAB (grant number 107754). ELW received funding from MRC Grant Reference MR/K012126/1. SAL was supported by the PANDORA-ID-NET Consortium (EDCTP Reg/Grant RIA2016E-1609). HM was supported by the Wellcome's Institutional Strategic Support Fund (grant number 204928/Z/16/Z).

*For correspondence: lawrencelby@gmail.com

**Competing interest:** The authors declare that no competing interests exist.

## Introduction

Lack of early infection-exposure has been associated with increased allergy-related disease (ARD) susceptibility later in life, the so-called 'hygiene hypothesis' (*Strachan, 1989*; *Bloomfield et al., 2006*; *Schaub et al., 2006*) or 'old friends hypothesis' (*Rook, 2010*; *Rook, 2011*). Studies, mainly in high-income countries (HICs), have highlighted the inverse relationship between pathogen-exposure and atopy, specifically for hepatitis A virus (*Strachan, 2000*), gut flora (*Björkstén et al., 1999*), intestinal parasites (*Yazdanbakhsh and Matricardi, 2004*), mycobacteria (*Shirakawa et al., 1997*), malaria (*Lell et al., 2001*), total burden of infections (*Martinez and Holt, 1999*; *Illi et al., 2001*), and farm animal sheds (*Loss et al., 2016*).

Despite increasing evidence indicating inverse relationships between pathogen-exposure and atopy, other studies suggest no association (*Jarvis et al., 2004*; *Cooper et al., 2003*) or even increased risk of atopy following a combination of early infections (*Seaton and Devereux, 2000*; *Bager et al., 2002*). Therefore, it remains unclear which infections, and at which times, possibly contribute to reducing the risk of atopy or ARDs.

There is a paucity of data from countries in tropical Africa where the infectious diseases burden remains high, yet prevalence of ARDs remains low, despite reasonably high prevalence of atopy. Similar to findings from HICs (*Bach, 2002*), we have shown previously that ARDs (eczema, wheeze, and rhinitis) decline with age, despite increasing atopy (*Lule et al., 2017*).

We therefore investigated the role of common early childhood infections in the development of atopy and ARDs, using data from the Entebbe Mother and Baby Study (EMaBS) birth cohort (*Webb et al., 2011*; *Elliott et al., 2007*). We hypothesised that (i) there are latent classes (LCs) of children with similar early infection-exposure and (ii) specific profiles of early infection-exposure are associated with reduced ARDs in later childhood.

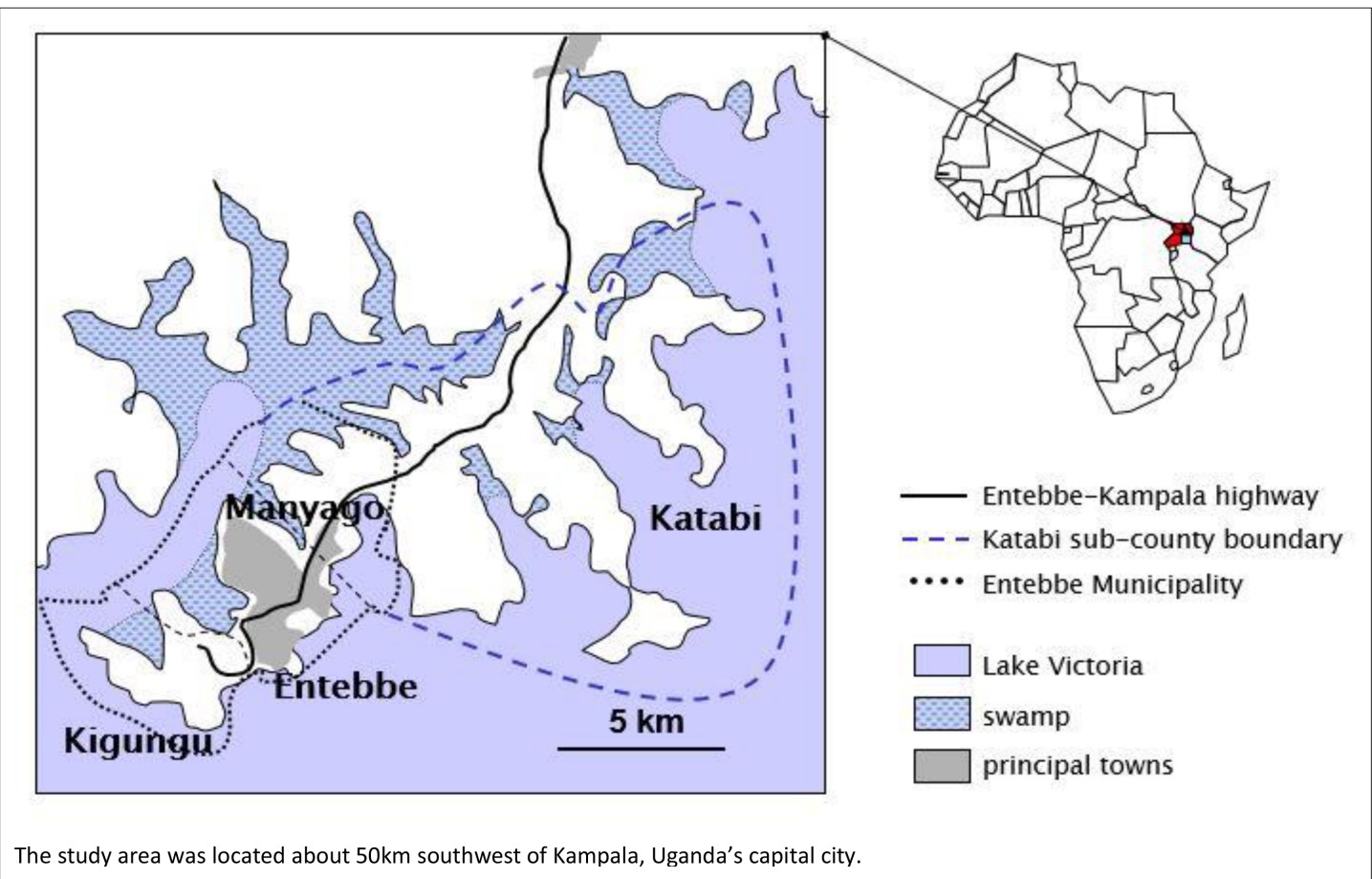

The study area was located about 50km southwest of Kampala, Uganda's capital city.

**Figure 1.** Map of Entebbe and Katabi showing the study area and the three main geographical zones of maternal residence at enrolment.

## Materials and methods

### Study design and population

The EMaBS has been described previously (*Elliott et al., 2007*). Briefly, the EMaBS, based at Entebbe General Hospital, Uganda, originated as a trial of anthelminthic treatment in pregnancy and early childhood, in a 2 × 2 (×2) factorial design. Between 2003 and 2005, women in their second or third trimester of pregnancy were randomised to single-dose albendazole (400 mg) vs. placebo and single-dose praziquantel (40 mg/kg) vs. placebo; their children were randomised to quarterly albendazole vs. placebo from age 15 months to 5 years (*Webb et al., 2011Elliott et al., 2007*). Cohort partici-pants continue under follow-up and had detailed data on ARDs collected at age 9 years. Our earlier study showed no effect of the anthelminthic intervention during pregnancy and early childhood on allergy-related outcomes at 9 years (*Namara et al., 2017*). The geography of the study area has been described previously (*Hillier et al., 2008*) the main zones of maternal residence at enrolment included Kigungu (a relatively isolated fishing village, at the tip of the Entebbe peninsula, n = 267), Entebbe (a commercial town, n = 1018), Manyago and Kabale (peri-urban, n = 687), and Katabi (comprised of peri-urban and rural areas, n = 486) (*Figure 1*).

### Ethical approval

The study was approved by ethics committees of the Uganda Virus Research Institute, London School of Hygiene and Tropical Medicine, and Uganda National Council for Science and Technology. Written informed consent and assent were obtained.

### Exposure variables

The main exposures for this study were children's most common clinical illnesses (malaria, diarrhoea, upper and lower respiratory tract infections [URTI and LRTI], intestinal helminths), and seroprevalence of herpes simplex virus (HSV), cytomegalovirus (CMV), and norovirus, in the first 5 years of life.

Malaria, diarrhoea, URTI, and LRTI were assessed and recorded by a doctor when a participant presented to the study clinic with an illness. Malaria was fever (temperature ≥37.5 °C) with para-sitaemia by thick film microscopy; diarrhoea was a child's carer's definition, with stool frequency recorded (*Webb et al., 2011*); URTI was the common cold while LRTI was cough, with difficulty in breathing, and fast breathing (defined by age), with or without abnormal breath sounds (*Elliott et al., 2007*).

At scheduled annual visits, a stool sample was collected and examined for the presence of helminth eggs using the Kato-Katz method (*Katz et al., 1972*). Two slides were examined for each sample, read within 30 min for hookworm and the following day for other worms including *Schistosoma mansoni*, *T. trichiura,* and *Ascaris lumbricoides* (*Mpairwe et al., 2014*).

Norovirus seropositivity was determined using immunoglobulin (Ig)G responses to human norovirus-like particles using enzyme-linked immunosorbent assays (ELISAs), as reported previously (*Thorne et al., 2018*). For 779 randomly selected children, plasma samples at 1 year were screened; any found negative were further tested at subsequent years until a positive response was detected (*Thorne et al., 2018*).

HSV and CMV seropositivity were determined using IgG responses, from standard commercially available ELISAs (DiaSorin, Saluggia, Italy). All plasma samples available at 2 and 5 years were assayed. Children whose samples were positive at 2 years had their 1-year samples tested. Those with nega-tives samples at 2 years were further assessed at years 3 and 4 until a positive response was detected.

### ARD outcomes and atopy at 9 years

ARDs and atopy at 9 years were the outcomes for this analysis. This is because there was a specific effort to collect data on ARDs, skin prick test (SPT) reactivity, and allergen-specific immunoglob-ulin E (asIgE) during the 9-year annual visit. We used the International Study on Allergy and Asthma in Children (ISAAC) questionnaire (*Asher et al., 1995*) to collect data on recent (last 12 months) reported wheeze, eczema (recurrent itchy rash with typical flexural distribution), and rhinitis, classified according to responses from caregivers for their children (*Namara et al., 2017*).

Atopy was assessed by SPT and asIgE as previously reported (*Lule et al., 2017*; *Namara et al., 2017*). SPT reactivity to *Dermatophagoides* mix (*D. farinae* and *D. pteronyssinus*), *Blomia tropicalis*, German cockroach, cat, mould, grass pollen, Bermuda grass, and peanut (ALK-Abelló, Laboratory

Specialities (Pty) Ltd, Randburg, South Africa) was assessed using standard methods (*Heinzerling et al., 2013*). A test was classified as positive if there was a papule of average size >3 mm, in the presence of saline (negative) and histamine (positive) controls. Plasma IgE to the commonest allergens in this setting (house dust mite [*D. pteronyssinus*] and German cockroach [*Blatella germanica*]) (*Mpairwe et al., 2008*) was measured using an in-house ELISA as previously described (*Mpairwe et al., 2011*; *Nkurunungi et al., 2018*). Because of the dynamic range of our in-house assay, it was not possible to use undiluted plasma, hence we optimised our assay to work with 20-fold diluted samples, with a lower detection limit of 15.625 ng/ml (equivalent to 312.5 ng/ml in undiluted plasma). We have previously shown that results from our ELISA were positively correlated with those from ImmunoCAP for both dust mite and cockroach (*Sanya et al., 2019*).

## Statistical analysis

Analyses aimed to classify participants into LCs based on infection-exposure during the first 5 years of life and to study the association between early infection-exposure and ARDs and atopy at 9 years. Since infection experience is an unobservable construct which can be inferred from multiple observed infections, we used the LC analysis (LCA) framework which has been shown to flexibly divide a population into mutually exclusive subgroups (*Goodman, 1974*; *Lanza et al., 2007*; *Nylund-Gibson and Choi, 2018*; *Velicer et al., 1996*; *Ryoo et al., 2018*).

We fitted LCA models using infection-exposure variables as indicators. Models were fitted, successively increasing the number of classes up to a seven-class solution. Two models were fitted: one considering infection-exposure during the first year of life (infancy) and the other using cumulative infection experience over the first 5 years, with no a priori restrictions to find consistent classes across the two models. First-year infection-exposure was considered because of the importance of infancy in immune development following exposure to several pathogens for the very first time. We used the Bayesian information criterion (BIC), adjusted BIC (ABIC), consistent Akaike information criterion (CAIC), and entropy to guide the choice of optimal number of classes (*Lanza et al., 2007*; *Swanson et al., 2012*).

We conducted multiple-group LCA (*Lanza et al., 2007*) to establish whether measurement invariance across sex holds. Multiple-group LCA allows class membership and item response probabilities to vary across groups. Formal tests for measurement invariance across sex were conducted for the selected models. We incorporated the covariates maternal area of residence, maternal history of asthma or eczema, and maternal and household socio-economic status (SES) at enrolment, to establish whether these impacted the probability of class membership, through multinomial logistic regression (*Lanza et al., 2007*). Maternal SES was determined by level of education, personal income, and occupation while household SES was based on building materials and the number of rooms and items owned (*Muhangi et al., 2007*). SES scores were split into two: low and high. We predicted ARDs and atopy at 9 years, from LC membership, using a SAS macro (*Dziak et al., 2017*), implementing a model-based approach as previously described (*Lanza et al., 2013*).

All models were estimated using SAS version 9.4 (SAS Institute Inc, Cary, NC) via PROC LCA which handles missing data on LC indicators under the missing at random assumption (*Lanza et al., 2007*).

## Results

Of 2345 live births in the EMaBS cohort, 2115 had data on illnesses during their first year of life, 1945 in the second, 1884 in the third, 1856 in the fourth, and 1838 in their fifth year (*Figure 2*).

Maternal and infant characteristics of the cohort participants, during the first 5 years of life, have been described previously (*Mpairwe et al., 2014*). Briefly, 52 % of the children were male, 90 % were of normal birth weight, 57 % were second to fourth born; maternal characteristics included 55 % with none/primary education, 3 % with history of asthma, 3 % with history of eczema, 44 % with hookworm infections, 21 % with *Mansonella perstans*, 18 % with *S. mansoni,* and 12 % with *Strongyloides stercoralis* (*Mpairwe et al., 2014*); 1214 children were seen at 9 years, 52 % of whom were male and 1179 had data collected on atopy and ARDs (*Lule et al., 2017*). Children seen at 9 years were similar to those not seen in terms of maternal marital status, BMI and worm infection at enrolment, and child's sex, eczema diagnosis (before age 1 year), and atopy (3 years) (*Lule et al., 2017*). Compared to mothers whose children did not attend the 9 -year visit, mothers of children seen at 9 years were

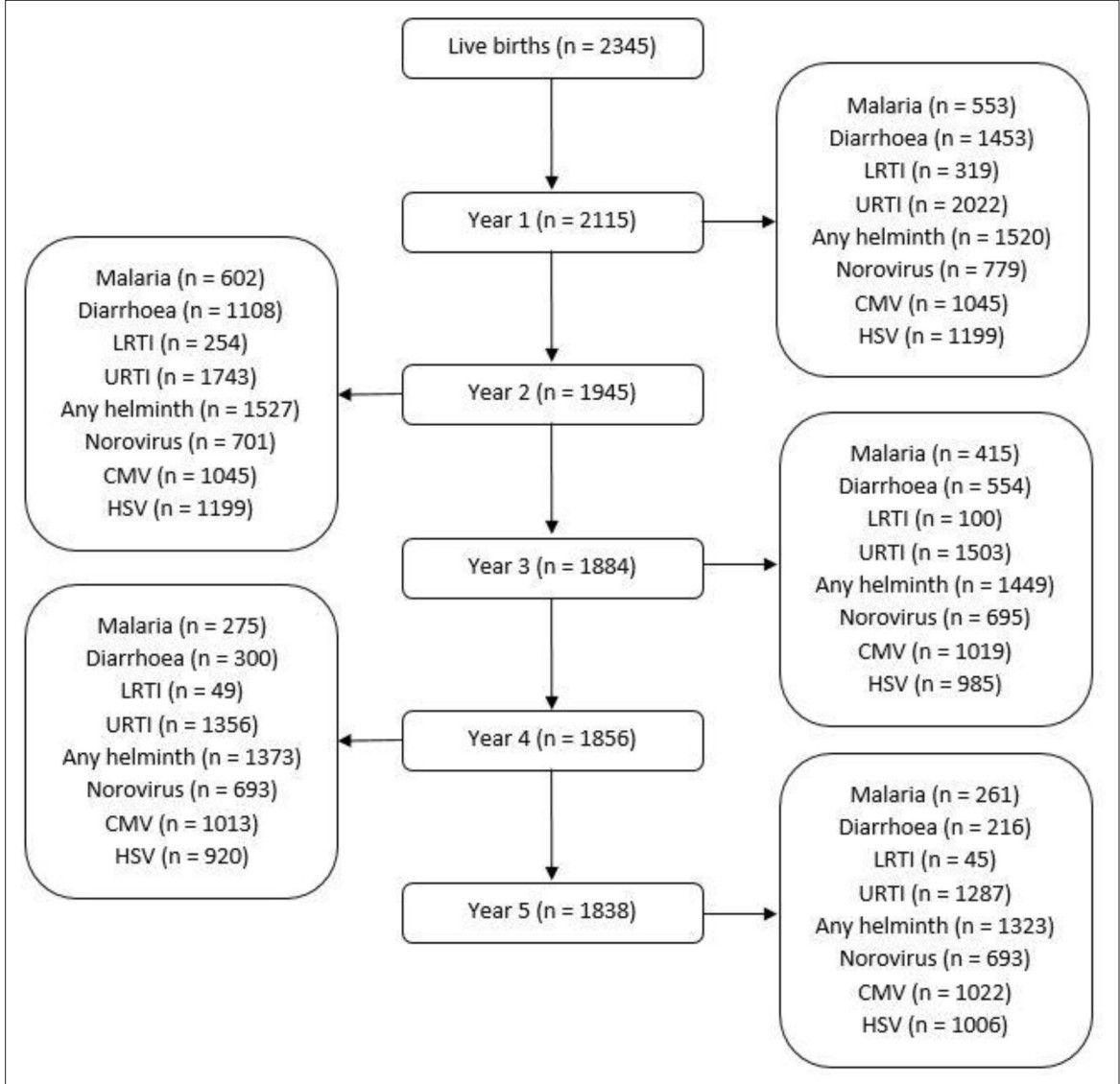

**Figure 2.** Flowchart showing number of the Entebbe Mother and Baby Study participants with data on illnesses/infections during each year. Numbers for each year (centre column of the flowchart) represent the children who presented at the study clinic at any point during that year. Numbers for malaria, diarrhoea, lower respiratory tract infection (LRTI) and upper respiratory tract infection (URTI) represent the children who were positively diagnosed (at least once during a given year) for each of those illnesses, on presentation at the study clinic. Numbers for any helminth, norovirus, cytomegalovirus (CMV), and herpes simplex virus (HSV) represent children samples assessed for any of those infections at the respective annual visit.

older, more likely to be Baganda (the tribe traditionally based in central Uganda), multiparous, and with higher SES at enrolment (*Lule et al., 2017*; *Namara et al., 2017*).

## Prevalence of childhood infections during the first five years of life

Diarrhoea and LRTI were common in the first year of life but declined by the fifth year, malaria was most common in the second year of life and then declined, URTI were consistently common throughout the first 5 years, helminth infections increased slightly, HSV increased steadily over the 5 years while CMV and norovirus seropositivity increased to over 85 % by the second year (*Figure 3*).

Considering cumulative experience of the various infections, over 95 % of the cohort participants had been infected with URTI, noroviruses, and CMV by the end of the third year of life. Diarrhoea infections had been reported by over 91 % by the third year increasing to 95 % by the fifth year. HSV increased steadily over time reaching a level of 91 % by the fifth year. The proportion of children with any recorded episode of malaria was 56% and 68% by the end of the third and fifth years, respectively.

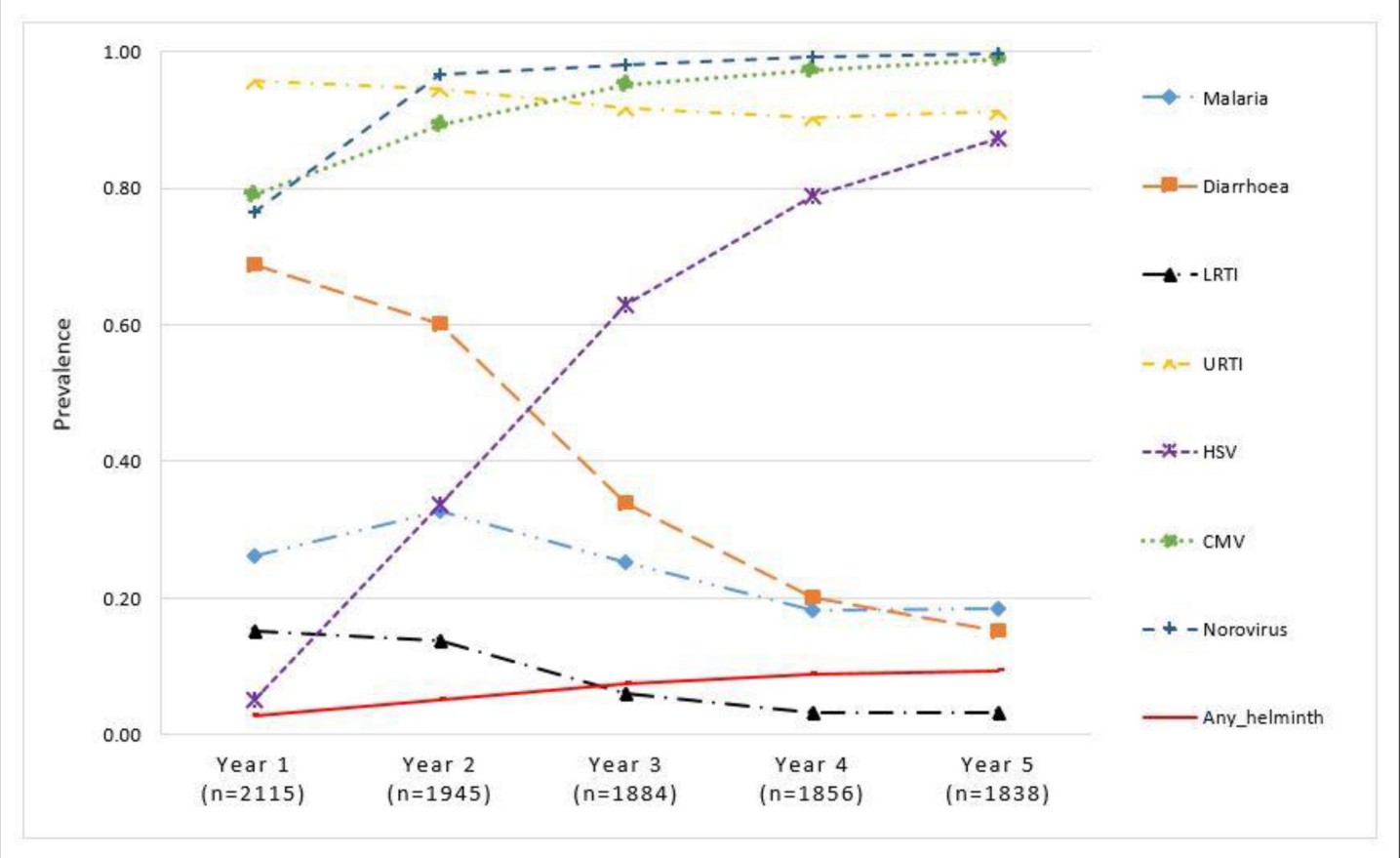

**Figure 3.** Proportion of children with illnesses/infections, over time, in the Entebbe Mother and Baby Study cohort.

LRTI increased gradually reaching 32% and 41% by the end of the third and fifth years, respectively. Helminth infections increased slowly reaching 31 % by the end of the fifth year. These cumulative figures are shown in *Figure 4*.

## LCA models

We conducted a sequence of models, considering additional classes gradually up to a seven-class solution. Models were fitted separately considering infection-exposure during the first year of life (infancy) and then using cumulative experience of infections over all the first 5 years. *Supplementary file 1a* summarises LCA results for the two different sets of models. Based on BIC, CAIC, and ABIC, two-class solution models were favoured for both first-year and 5-year cumulative infection experience. Although the entropy, which measures certainty in classifying latent statuses, favoured seven-class and six-class solutions for the two sets of models, we chose the two-class solutions for easy interpretation in addition to being favoured by other statistics.

Multiple-group LCA showed that measurement invariance by sex holds, based on the likelihood ratio difference test (p-value = 0.379 and p-value = 0.558 at year 1 and year 5, respectively) (*Supplementary file 1b*).

## Definition of LCs

*Figure 5* shows the probability of having been diagnosed with any of the infections, conditional on membership in either of the LCs, during the first year of life. The probabilities of membership in LC 1 and LC 2 are estimated as 32% and 68%, respectively. LC 1 was characterised by higher probabilities for malaria, diarrhoea, and LRTI (80% vs. 0.6%, 76 % vs. 65%, and 22% vs. 12 %, respectively) compared to LC 2. Conditional probabilities for other infections were similar between the LCs.

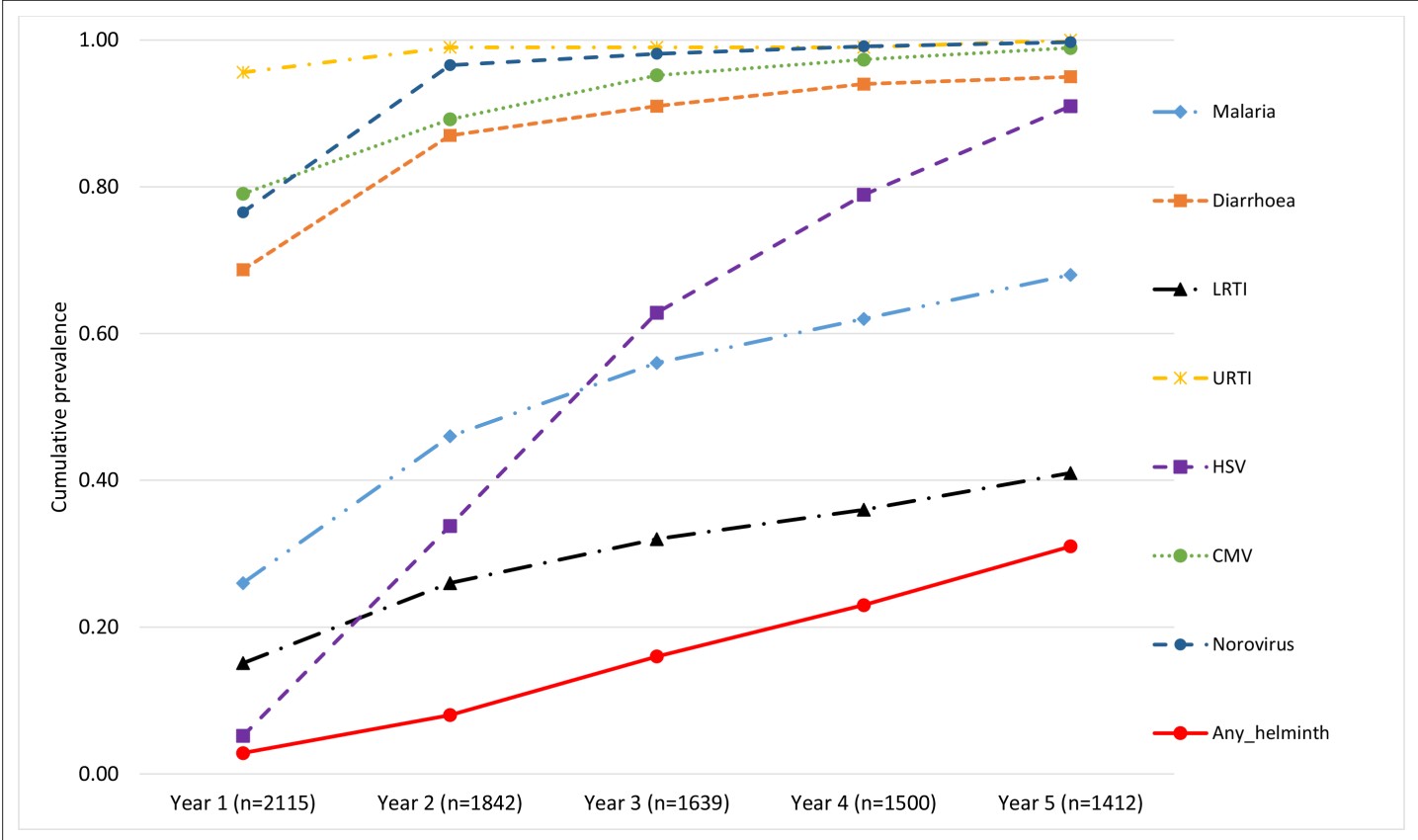

**Figure 4.** Cumulative infection experience, over time, in the Entebbe Mother and Baby Study cohort.

*Figure 6* shows the probability of having been diagnosed with any of the infections, conditional on membership in either of the LCs, considering cumulative infection prevalence over all the first 5 years of life. The probabilities of membership in LC 1 and LC 2 are estimated as 31% and 69%, respectively. LC 1 was characterised by higher probabilities for helminth infections, malaria, and LRTI (92% vs. 2.4%, 79 % vs. 64%, and 45% vs. 40 %, respectively) compared to LC 2. Probabilities for other infections were similar between the LCs.

We labelled the class with higher probabilities for various infections as LC 1 and the other as LC 2. Membership of LC 1 was 32 % in year 1 and 31 % in year 5 (considering cumulative infection experience) (*Table 1*).

## Effects of covariates on LC membership

At year 1, compared to children born in Entebbe town, those born in Kigungu fishing village were less likely to be in LC 1 [odds ratio (OR) = 0.33 (95% confidence interval (CI): 0.18, 0.62)], while those born in Katabi (peri-urban and rural areas) were more likely to be in LC 1 [OR = 2.43 (1.62, 3.65)] (*Table 2*).

Considering the LCA model for 5-year cumulative infection experience, children with higher maternal SES and those with higher household SES were less likely to be in LC 1 [OR = 0.51 (0.31, 0.82) and OR = 0.58 (0.25, 0.92), respectively] (*Table 2*). Children born in Kigungu and Katabi were more likely to be in LC 1 [OR = 5.73 (3.24, 8.12) for Kigungu and OR = 2.04 (1.07, 3.54) for Katabi] (*Table 2*). Maternal history of asthma or eczema was not associated with LC membership.

## Prevalence of atopy and ARD outcomes at 9 years

Prevalence of atopy and ARDs at 9 years has been reported elsewhere (*Lule et al., 2017*; *Namara et al., 2017*). Briefly, out of 1179 children for whom data on ARDs was collected at 9 years, prevalence of reported recent wheeze was 3.8%, eczema 4.9%, allergic rhinitis 4.6%, SPT positivity for at least one allergen 25%, for *Dermatophagoides* 18%, for *Blomia* 15%, for German cockroach 12%, for peanut

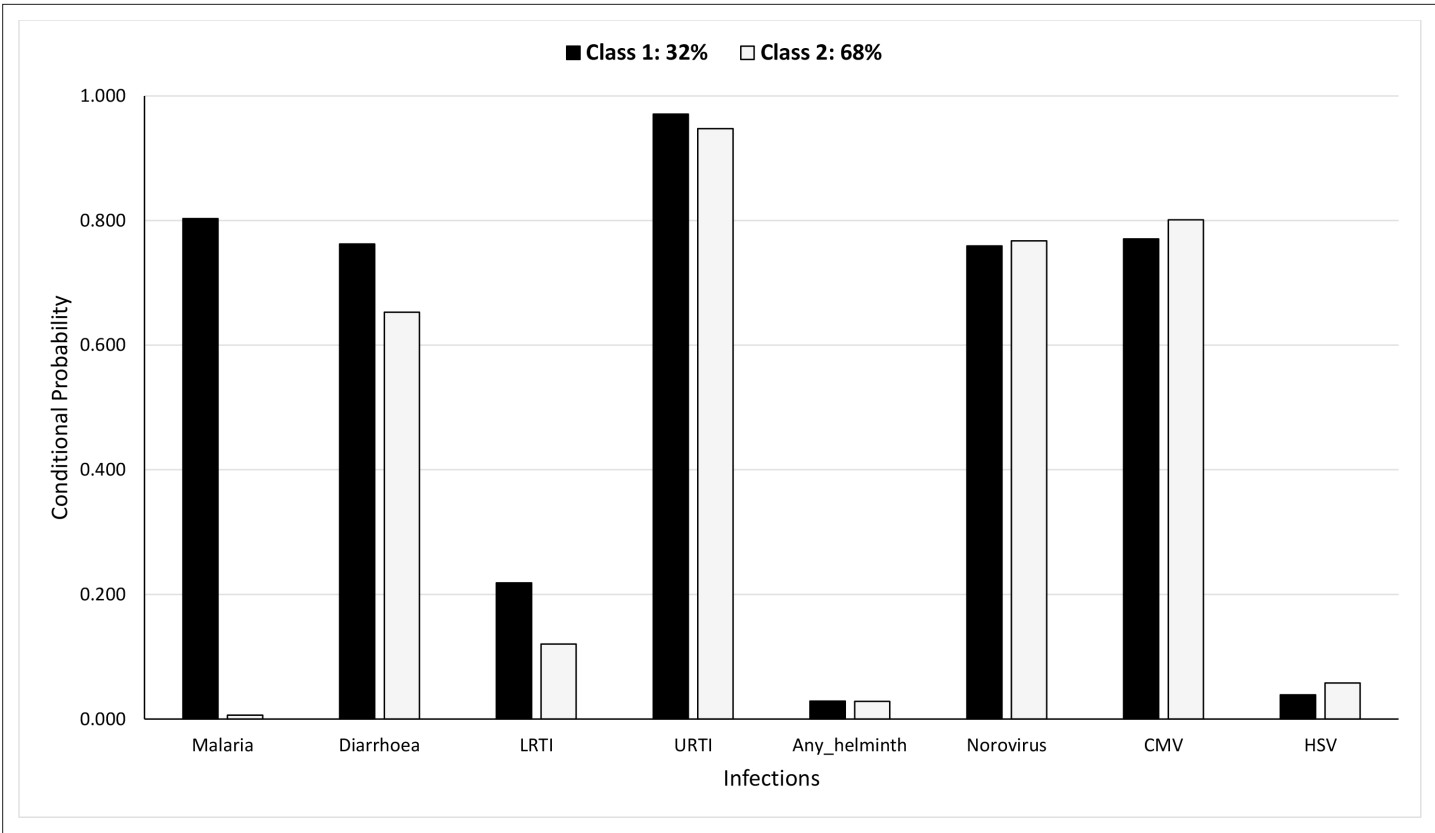

**Figure 5.** Probability of having infections conditional on latent class membership during the first year.

1.4%, for Bermuda grass 1.2%, for cat 1.1%, for pollen 0.9 %, and for mould 0.3% (*Lule et al., 2017*). Prevalence of detectable IgE to house dust mite was 29.4%, to cockroach 32.3 %, or to any of the two 44.1% (*Lule et al., 2017*; *Namara et al., 2017*).

## Associations between LC membership and atopy and ARD outcomes at 9 years

Based on covariate adjusted LCA of infection data collected during the first year of life, LC 1 membership was associated with lower probabilities of wheeze, eczema, rhinitis, and SPT positivity for *Dermatophagoides*; there was some suggestion of difference between the two LCs for SPT positivity overall and for *Blomia* (with SPT positivity less common for LC 1 children) but for cockroach and the IgE outcomes there was no suggestion of a difference (*Table 3*).

Estimates are adjusted for maternal area of residence, maternal history of asthma or eczema, and maternal and household SES at enrolment.

Based on covariate adjusted LCA of cumulative infection-exposure data collected throughout the first 5 years of life, LC 1 membership was associated with lower probabilities of SPT positivity for *Dermatophagoides* and for any of the other allergens (any of *Dermatophagoides* mix, *B. tropicalis*, German cockroach, cat, mould, grass pollen, Bermuda grass or peanut); there was some suggestion of difference between the two LCs for eczema but for SPT positivity for *Blomia,* or cockroach, wheeze, rhinitis, and the IgE outcomes there was no suggestion of a difference (*Table 4*).

Estimates are adjusted for maternal area of residence, maternal history of asthma or eczema, and maternal and household SES at enrolment.

## Discussion

In this study, using LCA, we identified two distinguishable groups of children based on either first-year or 5-year cumulative infection experience. In the first year of life, LC 1 was characterised by higher

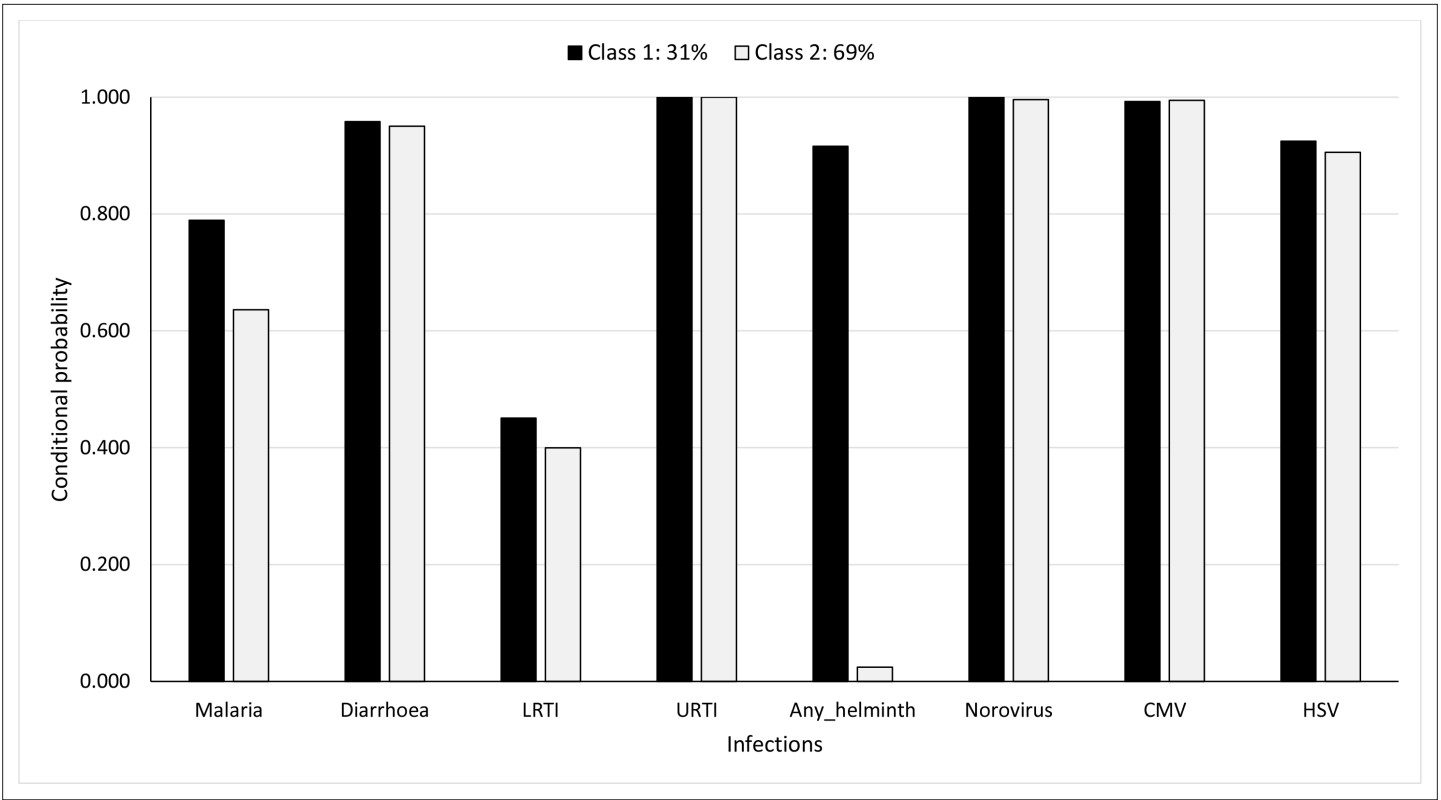

**Figure 6.** Probability of having infections conditional on latent class membership considering cumulative prevalence of infections over all the 5 years.

probabilities of malaria, diarrhoea, and LRTI, and membership of this class was associated with lower proportions of wheeze, eczema, rhinitis, and SPT positivity for *Dermatophagoides* at 9 years. Considering cumulative infection experience over the first 5 years of life, LC 1 was characterised by higher probabilities of helminth infections, malaria, and LRTI, and membership of this class was associated with lower probabilities of SPT positivity for *Dermatophagoides* and SPT positivity overall. Our results imply a significant contribution of immuno-modulating parasitic infections in early life to protection against the development of atopy and ARDs in later childhood in this tropical, low-income setting.

A major strength of this study lies in the use of LCA, a novel person-centred analysis approach, to characterise infection-exposure – an unobservable construct inferred from multiple observed infections in a given time period. At both time periods, class membership was characterised by differences in the burden of infections; malaria, diarrhoea, and LRTI during the first year and helminths, malaria, and LRTI during the 5-year period. LCA for first-year infection-exposure demonstrated the strong inverse association between infectious illnesses in infancy and ARDs in later childhood. On the other hand, LCA for cumulative infection experience over the first 5 years only illustrated strong inverse associations with atopy (SPT positivity). This suggests that infancy is a critical period during which the risk of ARDs in later childhood is established.

The finding that LC 1 membership, during the first year, was associated with lower proportions of wheeze, eczema, rhinitis, and SPT positivity for *Dermatophagoides* is consistent with observations from elsewhere (**Lell et al., 2001**) and with our earlier results which showed that children with more frequent clinical malaria infections in their first 5 years of life had a lower incidence of eczema in the same period (**Mpairwe et al., 2014**), and supports the hygiene hypothesis that

**Table 1.** Number (and proportion) of children in each latent class at year 1 and at year 5 considering cumulative infection experience.

| | LC 1 | LC 2 |
|---|---|---|
| | Number (%) | Number (%) |
| Year 1 (n = 2077) | 665 (32) | 1412 (68) |
| Year 5 (n = 1668) | 517 (31) | 1151 (69) |

LC is latent class. LC 1 represents the class with higher probabilities for various infections as compared to LC 2, however, the profile of infections changes at each time period.

**Table 2.** Odds ratio estimates for effects of covariates on membership in latent class 1 at year 1 and at year 5 considering cumulative infection experience.

| Covariate | Year 1 exposure | | 5 -Year cumulative exposure | |
|---|---|---|---|---|
| | Odds ratio | (95 % CI) | Odds ratio | (95% CI) |
| *Mother's social economic status* | | | | |
| Higher vs. lower | 0.77 | (0.58, 1.02) | **0.51** | **(0.31, 0.82)** |
| *Household's social economic status* | | | | |
| Higher vs. lower | 0.87 | (0.66, 1.14) | **0.58** | **(0.25, 0.92)** |
| *Maternal history of asthma or eczema* | | | | |
| History present vs. absent | 0.74 | (0.43, 1.27) | 0.50 | (0.24, 1.53) |
| *Area of residence at birth* | | | | |
| Kigungu vs. Entebbe | **0.33** | **(0.18, 0.62)** | **5.73** | **(3.24, 8.12)** |
| Manyago vs. Entebbe | 1.20 | (0.88, 1.63) | 1.08 | (0.88, 1.63) |
| Katabi vs. Entebbe | **2.43** | **(1.62, 3.65)** | **2.04** | **(1.07, 3.54)** |

CI is confidence interval. Bold values indicate statistically significant values at the 5% level.

infections in early life are associated with a lower risk of atopy and ARDs in later childhood (*Strachan, 2000*; *Björkstén et al., 1999*; *Yazdanbakhsh and Matricardi, 2004*; *Shirakawa et al., 1997*; *Lell et al., 2001*; *Martinez and Holt, 1999*; *Illi et al., 2001*; *Loss et al., 2016*).

We also describe the life-course of common infections in the first 5 years of life, in a tropical African birth cohort. We show that diarrhoea, malaria, and LRTI were more common in infancy but gradually declined by the fifth year, URTI remained very common while helminth infections increased gradually throughout the first 5 years, HSV increased steadily over time while CMV and norovirus infections were common throughout the 5 years. This result is especially important when studying which infections, occurring at which time, contribute to the development of atopy and ARDs. We show that in as much as two LCs are favoured at both time periods, conditional probabilities for the infections varied between first-year and 5-year cumulative exposure periods. Malaria was the largest contributor for LC 1 membership during the first year while helminth infection was the largest contributor for the 5-year period.

Compared to children born in Entebbe town, those born in Kigungu fishing village were less likely to be in LC 1 in year 1, while those born in Katabi (peri-urban and rural areas) were more likely to be in LC 1. These findings accord with our earlier study which showed comparable geographical variations in the prevalence of malaria in these areas (*Hillier et al., 2008*).

Another intriguing result was that based on 5-year cumulative infection-exposure, LC 1 was associated with a higher probability of helminth infections and membership in this class was associated with lower probabilities of SPT positivity, but was not associated with clinical ARDs. These findings accord with our earlier study which showed inverse associations between SPT positivity and *S. mansoni* infection in the rural setting (*Nkurunungi et al., 2019*).

A limitation of our study could be in the potential influence of missing data. The LCA models employ a maximum-likelihood routine which operates under a missing at random assumption. This assumption is inherently untestable, however, we consider it reasonable since the amount of missing data for each indicator variable, during each time period, was relatively low. Another limitation is that it was not possible to account for multiple exposure to infections (under the LCA approach) due to data on different infections being collected through different approaches. Some infections (HSV, norovirus, CMV, and helminths) were only captured (once, by serology) at the scheduled annual visits, while others (malaria, diarrhoea, LRTI, and URTI) were recorded every time a participant was presented to the study clinic. Another possible limitation could be in the use of 2077 and 1668 participants for determining LC membership in infancy, and between years 1 and 5, respectively, while only 1179 participants had outcome data at 9 years. In the analysis plan, which was developed before data analysis, we specified that we would use all available data to define LCs. We felt that this was appropriate

**Table 3.** Probability of allergy-related disease (ARD) outcomes and atopy at 9 years by latent class membership at year 1.

| ARD outcomes at 9 years | Latent class | Estimate | 95 % CI* Lower CI | Upper CI | p-Value |
|---|---|---|---|---|---|
| Wheeze | LC† 1 | 0.02 | 0.007 | 0.043 | **0.030** |
| | LC 2 | 0.05 | 0.038 | 0.071 | |
| Eczema | LC 1 | 0.03 | 0.013 | 0.056 | **0.043** |
| | LC 2 | 0.06 | 0.047 | 0.084 | |
| Rhinitis | LC 1 | 0.02 | 0.007 | 0.044 | **0.015** |
| | LC 2 | 0.06 | 0.045 | 0.081 | |
| SPT-any‡ | LC 1 | 0.22 | 0.178 | 0.278 | 0.201 |
| | LC 2 | 0.27 | 0.235 | 0.304 | |
| SPT-*Dermatophagoides* | LC 1 | 0.14 | 0.103 | 0.188 | **0.051** |
| | LC 2 | 0.20 | 0.171 | 0.232 | |
| SPT-cockroach | LC 1 | 0.11 | 0.074 | 0.149 | 0.760 |
| | LC 2 | 0.11 | 0.091 | 0.141 | |
| SPT-*Blomia* | LC 1 | 0.12 | 0.089 | 0.169 | 0.144 |
| | LC 2 | 0.16 | 0.138 | 0.196 | |
| IgE-*Dermatophagoides* | LC 1 | 0.26 | 0.214 | 0.322 | 0.261 |
| | LC 2 | 0.31 | 0.271 | 0.343 | |
| IgE-cockroach | LC 1 | 0.30 | 0.249 | 0.361 | 0.429 |
| | LC 2 | 0.33 | 0.295 | 0.370 | |
| IgE-any§ | LC 1 | 0.42 | 0.357 | 0.478 | 0.395 |
| | LC 2 | 0.45 | 0.411 | 0.490 | |

*CI is confidence interval.
†LC is latent class. LC 1 represents the class with higher probabilities for various infections as compared to LC 2.
‡SPT-any is skin prick test reactivity to any of *Dermatophagoides* mix, *Blomia tropicalis*, German cockroach, cat, mould, grass pollen, Bermuda grass, or peanut.
§IgE-any is allergen-specific plasma IgE (asIgE) to any of house dust mite (HDM, *Dermatophagoides pteronyssinus*) or German cockroach.

because it would describe the infection-exposure experience of the whole cohort, before going ahead to assess associations between this infection-exposure experience and the 9 -year outcomes. We had detailed data on important potential confounders which we adjusted for in this analysis, however we cannot rule out the possibility of residual confounding.

## Conclusion

In conclusion, we show that in this tropical African birth cohort children can be classified into two LCs based on their early childhood infection experience and that exposure to common infections, mainly malaria, during the first year of life, was associated with lower proportions of ARDs in later childhood. With cumulative infection experience over all the first 5 years, exposure to infections, mainly helminths, was associated with lower probabilities of SPT positivity in later childhood.

## Acknowledgements

We thank the study participants of the EMaBS, the staff of the Immunomodulation and Vaccines (I-Vac) Programme at the MRC/UVRI and LSHTM Uganda Research Unit, the midwives of the Entebbe

**Table 4.** Probability of allergy-related disease (ARD) outcomes and atopy at 9 years by latent class membership across the first 5 years.

| ARD outcomes at 9 years | Latent class | Estimate | 95 % CI* | | p-Value |
| | | | Lower CI | Upper CI | |
|---|---|---|---|---|---|
| Wheeze | LC† 1 | 0.02 | 0.009 | 0.065 | 0.272 |
| | LC 2 | 0.05 | 0.032 | 0.066 | |
| Eczema | LC 1 | 0.02 | 0.004 | 0.062 | **0.056** |
| | LC 2 | 0.07 | 0.050 | 0.090 | |
| Rhinitis | LC 1 | 0.02 | 0.008 | 0.067 | 0.139 |
| | LC 2 | 0.06 | 0.042 | 0.080 | |
| SPT-any‡ | LC 1 | 0.17 | 0.121 | 0.240 | **0.012** |
| | LC 2 | 0.29 | 0.249 | 0.325 | |
| SPT-*Dermatophagoides* | LC 1 | 0.11 | 0.069 | 0.169 | **0.010** |
| | LC 2 | 0.21 | 0.180 | 0.248 | |
| SPT-cockroach | LC 1 | 0.08 | 0.048 | 0.135 | 0.184 |
| | LC 2 | 0.12 | 0.099 | 0.154 | |
| SPT-*Blomia* | LC 1 | 0.11 | 0.070 | 0.169 | 0.096 |
| | LC 2 | 0.17 | 0.143 | 0.206 | |
| IgE-*Dermatophagoides* | LC 1 | 0.32 | 0.256 | 0.398 | 0.416 |
| | LC 2 | 0.28 | 0.247 | 0.325 | |
| IgE-cockroach | LC 1 | 0.34 | 0.271 | 0.414 | 0.658 |
| | LC 2 | 0.32 | 0.279 | 0.359 | |
| IgE-any§ | LC 1 | 0.48 | 0.403 | 0.555 | 0.316 |
| | LC 2 | 0.43 | 0.386 | 0.471 | |

*CI is confidence interval.

†LC is latent class. LC 1 represents the class with higher probabilities for various infections as compared to LC 2.

‡SPT-any is skin prick test reactivity to any of *Dermatophagoides* mix, *Blomia tropicalis*, German cockroach, cat, mould, grass pollen, Bermuda grass, or peanut.

§IgE-any is allergen-specific plasma IgE (asIgE) to any of house dust mite (HDM, *Dermatophagoides pteronyssinus*) or German cockroach.

General Hospital Maternity Department and the community field workers in Entebbe municipality and Katabi sub county.

## Additional information

### Funding

| Funder | Grant reference number | Author |
|---|---|---|
| African Academy of Sciences | DELTAS Africa Initiative SSACAB PhD Fellowship 107754 | Lawrence Lubyayi |
| Wellcome Trust | Senior Fellowship 064693 Senior Fellowship 079110 Senior Fellowship 95778 | Alison M Elliott |
| Medical Research Council | MR/K012126/1 | Emily L Webb |

| Funder | Grant reference number | Author |
|---|---|---|
| PANDORA-ID-NET Consortium | RIA2016E-1609 | Swaib A Lule |
| Wellcome Trust | Institutional Strategic Support 204928/Z/16/Z | Harriet Mpairwe |

The funders had no role in study design, data collection and interpretation, or the decision to submit the work for publication.

## Author contributions

Lawrence Lubyayi, Conceptualization, Data curation, Formal analysis, Methodology, Visualization, Writing - original draft, Writing – review and editing; Harriet Mpairwe, Gyaviira Nkurunungi, Swaib A Lule, Angela Nalwoga, Investigation, Writing – review and editing; Emily L Webb, Conceptualization, Investigation, Methodology, Supervision, Writing – review and editing; Jonathan Levin, Conceptualization, Methodology, Supervision, Writing – review and editing; Alison M Elliott, Conceptualization, Funding acquisition, Investigation, Supervision, Writing – review and editing

## Author ORCIDs

Lawrence Lubyayi http://orcid.org/0000-0001-5286-6488
Harriet Mpairwe http://orcid.org/0000-0003-1199-4859
Gyaviira Nkurunungi http://orcid.org/0000-0003-4062-9105
Alison M Elliott http://orcid.org/0000-0003-2818-9549

## Ethics

Human subjects: Parents or guardians of the children provided written informed consent, and children eight years or older provided written informed assent. This consent was to participate in the study, and to publish anonymous results. The study was approved by ethics committees of the Uganda Virus Research Institute [reference number GC/127], London School of Hygiene and Tropical Medicine [application number 790] and Uganda National Council for Science and Technology [reference number MV 625].

## Decision letter and Author response

Decision letter https://doi.org/10.7554/eLife.66022.sa1
Author response https://doi.org/10.7554/eLife.66022.sa2

# Additional files

## Supplementary files

• Supplementary file 1. Additional results tables for model fit statistics and measurement invariance for year 1 and year 5 cumulative infection experience. (a) Model fit statistics for latent class analysis at year 1 and at year 5 considering cumulative infection experience. (b) Results for measurement invariance by sex, at year 1 and at year 5.
• Transparent reporting form
• Reporting standard 1. STROBE checklist.

## Data availability

Data is available on request via https://doi.org/10.17037/DATA.00002438. To gain access to the data please complete the application process via the website. Requests will be reviewed and assessed by the corresponding author, in consultation with the LSHTM's Research Data Manager and relevant LSHTM staff members responsible for research governance and data protection. Applications will be evaluated on the basis of their compatibility with the study's research objectives and the ability to provide de-identified data sufficient to meet the intended purpose, without breaching participant confidentiality or the study's ethical and legal commitments. Successful applicants will be asked to sign a Data Transfer Agreement prior to being provided with the data.

The following dataset was generated:

| Author(s) | Year | Dataset title | Dataset URL | Database and Identifier |
|---|---|---|---|---|
| Lubyayi L, Webb E | 2021 | EMaBS 1 to 5 years illness data and ARD outcomes at 9 years | https://doi.org/10.17037/DATA.00002438 | London School of Hygiene & Tropical Medicine (LSHTM) Data Compass, 10.17037/DATA.00002438 |

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
