## [Decision Letter]

**Acceptance summary:**

The authors present an analysis of the potential effects of early life infections on allergy related outcomes at 9 years of age. To do this, they use large study population from Entebbe, Uganda, that was the subject of randomized controlled trails of the effects of anthelmintic treatment given during pregnancy to mothers and later to children during up to 5 years of age. The authors use these data to do a cohort analysis of the effects of early life infections on allergy outcomes later in childhood. What makes this study important and extremely unusual are the wide range of infectious diseases exposure measurements taken and the detailed data on allergic outcomes at 9 years.

**Decision letter after peer review:**

Thank you for submitting your article "Infection-exposure in infancy is associated with reduced allergy-related disease in later childhood in a Ugandan cohort" for consideration by *eLife*. Your article has been reviewed by 3 peer reviewers, and the evaluation has been overseen by a Reviewing Editor and Jos van der Meer as the Senior Editor. The reviewers have opted to remain anonymous.

The reviewers have discussed their reviews with one another, and we have drafted this to help you prepare a revised submission.

Essential revisions:

The authors investigated the association between infection exposure in early life and susceptibility to allergy-related diseases (ARD) in the Entebbe Mother and Baby Study (EMaBS) birth cohort, using the latent class analysis (LTA). Most common clinical illnesses such as malaria, diarrhea, upper and lower respiratory tract infections, intestinal helminths, herpes simplex virus infection, cytomegalovirus and norovirus infections are analyzed in the first 5 years of life, as well as ARD through skin prick test and allergen-specific Immunoglobulin E (asIgE) at 9 years of age. At each year of life, children were divided into two groups (latent classes) defined by their infection-exposure history in that year and probabilities of ARDs outcomes in these groups were compared. Authors concluded that early childhood infection-exposure to malaria and diarrhoea was associated with lower prevalence of ARDs in later childhood.

Strengths:

1. The cohort with a large number of patients originating from 4 different geographical areas and followed throughout the first 5 years of age, with a detailed follow-up at 9 years of age.

2. The amount of data obtained and available for each patient, which allows an extended analysis and comparison.

Weaknesses:

The major weaknesses of the paper concern the statistics applied.

1. Analysis of the exposure: It is not the best approach to define exposure classes for each year and separately analyze their associations with outcomes at 9 years. Outcomes could be affected by any/all exposures in years 1-5, so analysis of a single year is likely to be confounded by exposure in the other years and provides biased estimates. Additionally, many children were likely to be exposed to some infections multiple numbers of time. It is unclear how this was handled in the analysis. Not accounting for the multiple exposure could also bias the results as, under the study hypothesis, children with multiple exposure would be those mostly protected from the ARDs.

2. Analysis of the association with outcome: The association between exposure and ARDs is studied in separated from other possible risk factors for ARDs, such as genetic atopic predisposition, early childhood allergen exposure and sensitization, maternal smoking during pregnancy, poor dietary factors, lack of breast-feeding, childhood obesity, having a certain immunologic predisposition, air pollution, and frequent immunizations in childhood. These risk factors as well as ARDs status at 1-5 years are crucial in understanding the association between infection exposure and development of ARDs. This needs to be discussed.

3. Interpretation of results: Possible bias in results aside, they are very difficult to interpret. Results of the analyses are inconsistent across years. Differences between classes with respect to exposure patterns and probabilities of outcomes are rather small, therefore it is impossible to draw any sound conclusions on the association (or lack of it) between exposures and outcomes. Authors' strong conclusion is in contradiction to the above, but also to their own assessment of weaknesses of the study presented in the discussion. The interpretation of the results was primary based on p-values, when multiple testing was certainly an issue with 10 outcomes and repeated analysis for each of 5 years.

As the statistical analysis is considered the major weakness these issues are listed first.

1. We think that the data should have been analyzed together, not split by year. We do not understand why single LCA exploring exposure over 5 years could not be performed. We think LTA, which authors were planning to conduct in the first place, would not be the right approach as it explores changes in the exposure over time – while the objective of this analysis was to relate the exposure in the first years to ARDs at nine years of age. Latent classes defined over 5 years would account for any changes in exposures.

2. The standard procedure for conducting LCA is to conduct a sequence of models, starting with a one-class model and then specifying models with one additional class at a time. Restriction of the analysis to {greater than or equal to}2 classes is very likely to produce some "significant" associations, especially when multiple testing is performed.

3. The analysis cohort is not described at all, which is important especially in the context of risk factors for ARDs and the history of ARDs.

4. Exposure to an infection is presented as a proportion of children with that infection per year, but not presented per child. For each infection, what was the incidence rate and its variability between children? It is unclear how exposure was captured for each child in the analysis – as an incidence or as `at least one infection'? Incidence could have been a better representation of the exposure.

5 Other risk factors for ARDs should have been considered in the analysis otherwise the association between exposure and outcome could be seriously biased.

6. It would be worth summarizing the effects of the RCTs on allergy outcomes to 9 years in the introduction, so that the reader is informed of any potential effects of treatment arms on study outcomes, given that the analysis does not take into account the original design. It might also be worth doing sub-group analyses among control-arms of both RCTs to see if the results of those, albeit likely underpowered analyses, are consistent with the whole-group analysis such that treatment allocations can be safely ignored. It would be useful to provide some discussion in the limitations section of the potential effects, if any, of these RCTs on study findings.

7. The cohort is composed of 2345 children, but only 1179 have data on ARD at nine years of age. As the goal is to investigate a possible association between infection exposure in early life and susceptibility to ARDs, shouldn't the number of analyzed subjects be 1179 in total? The authors should either repeat the LTA or discuss the difference in population samples (2345 vs 1179) as a weakness/limitation of their study.

8. The authors describe the ARD information methodology and state (page 6, lines 122-124) that they used 20-fold diluted plasma, thus significantly reducing the sensitivity of their test. IgE tests are normally done on undiluted serum. The reason for diluting plasma 20x should be given. Moreover, information concerning validation of this assay against a gold standard (e.g. immunoCAP, TFS) should be at least mentioned here (just a reference is not enough). Finally, what SPT wheal (not papule!) diameter corresponds to the threshold of 312.5 ng/ml? Is the IgE test sensitive enough?

9. Plasma samples were only tested against Dermatophagoides pteronyssinus and Blatella germanica. Please give information why this choice has been made. Why were only these two allergen extract examined with IgE assays?

10. Please insert more information to help the interpretation of data (the favored models) of table 1 (page 11) and consider to move the table in an electronic repository, considering that this table is occupying one page and the respective information is not the major focus of the paper.

11. It seems that children from Entebbe were compared against the other 3 areas, but these were not compared among themselves. Please provide a brief explanation for this choice.

12. The LCA model is fitted separately at each time point: 1-5 years of age. This provides a lot of information, but can also originate some confusion: children born in Kigungu are less likely to be in LC 1 – and therefore have a higher probability for various infections – when they are 2 years of age, but the opposite occurs when they are 4 years of age (page 14, lines 242-247). This age dependency of LTA categorization needs to be discussed: is it a limitation of the study?

13. SPT or IgE positivity is not equivalent of ARD. Please reformulate the definition of atopic and/or allergic outcome at 9y of age in the whole paper.

14. A table showing the prevalence of SPT/IgE positivity for each allergen extract should be given somewhere in the results. In particular, only a few extracts (blomia, mites and cockroach) are used in the presentation of the table 4. However, many more allergen extracts have been examined. Probably, the prevalence of e.g. pollen sensitization was low. However, this is not shown in the paper.

*Reviewer #1:*

The authors present an analysis of the potential effects of early life infections on allergy and allergic diseases at 9 years of age. To do this, they use large study population from Entebbe, Uganda, that was the subject of previous randomized intervention studies. The authors use these data to do a cohort analysis of the effects of early life infections on allergy later in childhood. What makes this study important and extremely unusual are the wide range of infectious diseases exposure measurements taken and the detailed data on allergic outcomes at 9 years. Because many of these infections are strongly associated, the authors used a statistical technique to infer how underlying patterns of infectious diseases might affect later allergy. The authors show that having more types of different infections during the first year of life appears to protect the children from having allergy at school-age.

*Reviewer #2:*

The authors investigate the association between infection exposure in early life and susceptibility to allergy-related diseases (ARD) in the Entebbe Mother and Baby Study (EMaBS) birth cohort, using the latent class analysis (LTA). Most common clinical illnesses such as malaria, diarrhoea, Upper and Lower Respiratory Tract Infections, intestinal helminths, Herpes Simplex Virus, Cytomegalovirus and norovirus are analysed in the first 5 years of life, as well as ARD through skin prick test and allergen-specific Immunoglobulin E (asIgE) at 9 years of age. Using LTA, children were allocated into two latent classes: LC1 and LC2, being LC1 the one with higher probabilities for various infections. The Authors concluded that early childhood infection-exposure is associated with lower prevalence of ARDs in later childhood.

General comments: This article provides interesting new data within a topic of high interest and importance.

Strengths:

– The cohort, with a large amount of patients originating from 4 different geographical areas and followed throughout the first 5 years of age, with a detailed follow-up at 9 years of age.

– The amount of data obtained and available for each patient, which allows an extended analysis and comparison, therefore leading to new knowledge and further hypothesis.

– The outcome of the work.

*Reviewer #3:*

Lubyayi, Lawrence et al., conducted a secondary data analysis to explore association between early childhood infections and development of atopy and allergy related diseases (ARDs) in later childhood. Data from the Entebbe Mother and Baby Study birth cohort were used. Information on exposure to children's most common clinical illnesses and seroprevalence of common viruses was available in the first five years of life; allergy-related disease outcomes at nine years of age were collected using allergy tests and standard questionnaires. Multiple latent class analyses were conducted. At each year of life, children were divided into two groups (latent classes) defined by their infection-exposure history in that year and probabilities of ARDs outcomes in these groups were compared. Authors concluded that early childhood infection-exposure to malaria and diarrhoea was associated with lower prevalence of ARDs in later childhood.

The study has a number of serious weaknesses:

1. Analysis of the exposure: It is not the best approach to define exposure classes for each year and separately analyse their associations with outcomes at 9 years. Outcomes could be affected by any/all exposures in years 1-5, so analysis of a single year is likely to be confounded by exposure in the other years and provides biased estimates. Additionally, many children were likely to be exposed to some infections multiple number of times. It is unclear how this was handled in the analysis. Not accounting for the multiple exposure could also bias the results as, under the study hypothesis, children with multiple exposure would be those mostly protected from the ARDs.

2. Analysis of the association with outcome: The association between exposure and ARDs is studied in a complete separation from other possible risk factors for ARDs, such as genetic atopic predisposition, early childhood allergen exposure and sensitization, maternal smoking during pregnancy, poor dietary factors, lack of breast-feeding, childhood obesity, having a certain immunologic predisposition, air pollution, and frequent immunizations in childhood. These risk factors as well as ARDs status at 1-5 years are crucial in understanding the association between infection exposure and development of ARDs.

3. Interpretation of results: Possible bias in results aside, they are very difficult to interpret. Results of the analyses are inconsistent across years. Differences between classes with respect to exposure patterns and probabilities of outcomes are rather small, therefore it is impossible to draw any sound conclusions on the association (or lack of it) between exposures and outcomes. Authors' strong conclusion is in contradiction to the above, but also to their own assessment of weaknesses of the study presented in the discussion. The interpretation of the results was primary based on p-values, when multiple testing was certainly an issue with 10 outcomes and repeated analysis for each of 5 years.

---

## [Author Response]

Essential revisions:The authors investigated the association between infection exposure in early life and susceptibility to allergy-related diseases (ARD) in the Entebbe Mother and Baby Study (EMaBS) birth cohort, using the latent class analysis (LTA). Most common clinical illnesses such as malaria, diarrhea, upper and lower respiratory tract infections, intestinal helminths, herpes simplex virus infection, cytomegalovirus and norovirus infections are analyzed in the first 5 years of life, as well as ARD through skin prick test and allergen-specific Immunoglobulin E (asIgE) at 9 years of age. At each year of life, children were divided into two groups (latent classes) defined by their infection-exposure history in that year and probabilities of ARDs outcomes in these groups were compared. Authors concluded that early childhood infection-exposure to malaria and diarrhoea was associated with lower prevalence of ARDs in later childhood.Strengths:1. The cohort with a large number of patients originating from 4 different geographical areas and followed throughout the first 5 years of age, with a detailed follow-up at 9 years of age.2. The amount of data obtained and available for each patient, which allows an extended analysis and comparison.Weaknesses:The major weaknesses of the paper concern the statistics applied.1. Analysis of the exposure: It is not the best approach to define exposure classes for each year and separately analyze their associations with outcomes at 9 years. Outcomes could be affected by any/all exposures in years 1-5, so analysis of a single year is likely to be confounded by exposure in the other years and provides biased estimates. Additionally, many children were likely to be exposed to some infections multiple numbers of time. It is unclear how this was handled in the analysis. Not accounting for the multiple exposure could also bias the results as, under the study hypothesis, children with multiple exposure would be those mostly protected from the ARDs.

We thank the reviewers for this comment. We have now analysed exposure over 5 years altogether and present these findings in the updated manuscript. We also report results for first year exposure, considering the importance of infancy in immune development following exposure to several pathogens for the very first time.

Accounting for multiple exposure to infections in the study participants is challenging under the LCA approach. Since we are using several indicator infections with some only captured based on data from the scheduled annual visits (HSV, norovirus, CMV and helminths), while others were recorded every time a participant was presented to the study clinic (malaria, diarrhoea, LRTI and URTI), it was not possible to account for multiplicity for some and not for others in a consistent way. We have added this as a potential limitation in the Discussion section (lines 450 – 455).

2. Analysis of the association with outcome: The association between exposure and ARDs is studied in separated from other possible risk factors for ARDs, such as genetic atopic predisposition, early childhood allergen exposure and sensitization, maternal smoking during pregnancy, poor dietary factors, lack of breast-feeding, childhood obesity, having a certain immunologic predisposition, air pollution, and frequent immunizations in childhood. These risk factors as well as ARDs status at 1-5 years are crucial in understanding the association between infection exposure and development of ARDs. This needs to be discussed.

We did not analyse associations between exposure and ARDs separately from other risk factors. The LCA approach allows for incorporation of covariates by way of establishing the effect of covariates on latent class membership. In our original analysis, we incorporated the covariates maternal area of residence, and maternal and household socio-economic status (SES) at enrolment. We have now additionally included maternal history of asthma and eczema as a proxy for genetic atopic predisposition. We appreciate that the other factors highlighted by the reviewer could be relevant, however, we did not have data on genetic atopic predisposition (other than as maternal history), early childhood allergen sensitisation was only assessed for a subset of participants at 3 years, and some factors (e.g. maternal smoking, lack of breast-feeding, childhood obesity) were universally uncommon in this setting, other factors (e.g. childhood immunisations which were conducted at the study clinic) were universally common. Air pollution may have differed by maternal area of residence, which was considered, but was not otherwise assessed. Effects of covariates on LC membership are highlighted in the manuscript (lines 299 – 309).

3. Interpretation of results: Possible bias in results aside, they are very difficult to interpret. Results of the analyses are inconsistent across years. Differences between classes with respect to exposure patterns and probabilities of outcomes are rather small, therefore it is impossible to draw any sound conclusions on the association (or lack of it) between exposures and outcomes. Authors' strong conclusion is in contradiction to the above, but also to their own assessment of weaknesses of the study presented in the discussion. The interpretation of the results was primary based on p-values, when multiple testing was certainly an issue with 10 outcomes and repeated analysis for each of 5 years.

We thank the reviewers for this comment. We believe that since we have now re-analysed the exposure data over 5 years, the results should now be simpler to interpret.

Regarding the issue of multiple testing, there are fewer tests now since we have restricted the analysis to only the first year and then cumulative infection experience over the five years. Secondly, we employed the LCA approach because it helps in dealing with the issue of multiple related exposures. Thirdly, regarding the 10 atopy and ARD outcomes, we think that it would be overly conservative if we consider adjustment for multiplicity because these outcomes are not a random collection, they were selected because there is prior evidence for their associations with infection exposure and they are mostly correlated with each other.

As the statistical analysis is considered the major weakness these issues are listed first.1. We think that the data should have been analyzed together, not split by year. We do not understand why single LCA exploring exposure over 5 years could not be performed. We think LTA, which authors were planning to conduct in the first place, would not be the right approach as it explores changes in the exposure over time – while the objective of this analysis was to relate the exposure in the first years to ARDs at nine years of age. Latent classes defined over 5 years would account for any changes in exposures.

We thank the reviewers for this comment. We have now analysed exposure over 5 years altogether. We also report results for first year LCA, considering the importance of infancy in immune development following exposure to several pathogens for the very first time. We have also removed all the text relating to LTA from the paper.

2. The standard procedure for conducting LCA is to conduct a sequence of models, starting with a one-class model and then specifying models with one additional class at a time. Restriction of the analysis to {greater than or equal to}2 classes is very likely to produce some "significant" associations, especially when multiple testing is performed.

We thank the reviewers for this comment. Indeed, we conducted a sequence of models, considering additional classes gradually and using Bayesian Information Criterion (BIC), adjusted BIC (ABIC), consistent Akaike Information Criterion (CAIC) and entropy to guide the choice of optimal number of classes (without reference to the outcome variables). We considered models with 2 classes or more because we needed to account for the effects of covariates on latent class membership, otherwise this is not possible with one-class models which again would not require the use of LCA analysis.

3. The analysis cohort is not described at all, which is important especially in the context of risk factors for ARDs and the history of ARDs.

We thank the reviewers for this comment. We have now included more information about the analysis cohort (lines 198 – 209).

4. Exposure to an infection is presented as a proportion of children with that infection per year, but not presented per child. For each infection, what was the incidence rate and its variability between children? It is unclear how exposure was captured for each child in the analysis – as an incidence or as `at least one infection'? Incidence could have been a better representation of the exposure.

We describe exposure as at least one infection reported to the study clinic during the given period for malaria, diarrhoea, LRTI and URTI. For HSV, CMV, Norovirus and any helminths infections, samples collected at the annual visits were assessed. The age of seroconversion was determined for HSV, CMV and norovirus. Exposure to helminth infections was determined based on testing positive at any annual visit. As such, we could not use incidence for representation of the exposure, hence the use of prevalence estimates indicating cumulative infection at any time during the five years.

5 Other risk factors for ARDs should have been considered in the analysis otherwise the association between exposure and outcome could be seriously biased.

The LCA approach allows for incorporation of covariates by way of establishing the effect of covariates on latent class membership. In our original analysis, we incorporated the covariates maternal area of residence, and maternal and household socio-economic status (SES) at enrolment. We have now additionally included maternal history of asthma and eczema as a proxy for genetic atopic predisposition. Indeed, inclusion of covariates in these models leads to different results as compared to models without covariates; we cannot rule out the possibility of residual confounding, which we have now highlighted in the discussion (lines 462 – 463).

6. It would be worth summarizing the effects of the RCTs on allergy outcomes to 9 years in the introduction, so that the reader is informed of any potential effects of treatment arms on study outcomes, given that the analysis does not take into account the original design. It might also be worth doing sub-group analyses among control-arms of both RCTs to see if the results of those, albeit likely underpowered analyses, are consistent with the whole-group analysis such that treatment allocations can be safely ignored. It would be useful to provide some discussion in the limitations section of the potential effects, if any, of these RCTs on study findings.

Thank you for this comment. The effects of the RCT on allergy outcomes at 9 years were described in our earlier report (Namara et al., Pediatric Allergy and Immunology 2017; 28(8): 784-792). We have now briefly mentioned these results in this manuscript (lines 75 – 76). We can confirm that because treatment allocation had no effect on the outcomes at nine years, it can be safely ignored for this paper.

7. The cohort is composed of 2345 children, but only 1179 have data on ARD at nine years of age. As the goal is to investigate a possible association between infection exposure in early life and susceptibility to ARDs, shouldn't the number of analyzed subjects be 1179 in total? The authors should either repeat the LTA or discuss the difference in population samples (2345 vs 1179) as a weakness/limitation of their study.

Thank you for this comment. In the analysis plan which was developed before data analysis, we specified that we would use all available data to define latent classes. We felt that this was appropriate because it would describe the infection exposure experience of the whole cohort, before going ahead to assess associations between this infection-exposure experience and the 9 year outcomes. We have included a few sentences in the Discussion section discussing this and whether it could have introduced bias (lines 455 – 461). Here we have referenced two earlier papers which reported that characteristics of those seen at 9 years were similar to those not seen in terms of maternal marital status, BMI and worm infection at enrolment, and child's sex, eczema diagnosis (before age 1 year) and atopy (at 3 years) (lines 204 – 209).

8. The authors describe the ARD information methodology and state (page 6, lines 122-124) that they used 20-fold diluted plasma, thus significantly reducing the sensitivity of their test. IgE tests are normally done on undiluted serum. The reason for diluting plasma 20x should be given. Moreover, information concerning validation of this assay against a gold standard (e.g. immunoCAP, TFS) should be at least mentioned here (just a reference is not enough). Finally, what SPT wheal (not papule!) diameter corresponds to the threshold of 312.5 ng/ml? Is the IgE test sensitive enough?

It is true that standard IgE tests such as ImmunoCAP normally use undiluted serum/plasma. For logistical reasons, we used an in-house ELISA assay to detect IgE, using native IgE from human myeloma plasma as a standard. Because of the dynamic range of our in-house assay, it was not possible to use undiluted plasma, hence we optimised our assay to work with 20-fold diluted samples, with a lower detection limit of 15.625 ng/ml. As our assay is not a standard assay, we do not use it to classify sensitised vs non-sensitised individuals, rather, we use it to define detectable vs undetectable IgE levels. We also do not use it to determine what SPT wheal size corresponds to the lower detection limit. We have previously shown that results from our in-house ELISA were positively correlated with ImmunoCAP results for both dust mite and cockroach (Sanya et al., Clinical infectious diseases 2019; 68(10): 1665-1674). We have now added some of the above explanation to the manuscript text (lines 129 – 133).

9. Plasma samples were only tested against Dermatophagoides pteronyssinus and Blatella germanica. Please give information why this choice has been made. Why were only these two allergen extract examined with IgE assays?

We have previously shown that house dust mites (Dermatophagoides spp, Blomia tropicalis) and cockroach allergens are the commonest in this study setting (Mpairwe et al., Trans R Soc Trop Med Hyg. 2008;102(4):367-37). Our recent studies using the ISAC allergen microarray confirm this (Nkurunungi et al., Allergy. 2020;76(1):233-246.) Therefore, taking into account the budgetary constraints, we chose to test dust mite- and cockroach-specific IgE. We have now added this further explanation to the manuscript text (lines 126 – 127).

10. Please insert more information to help the interpretation of data (the favored models) of table 1 (page 11) and consider to move the table in an electronic repository, considering that this table is occupying one page and the respective information is not the major focus of the paper.

We thank the reviewers for this comment. We have now moved the former table 1 to supplementary material (Supplementary table S1). We have also added some more information to aid the interpretation of this result (lines 236 – 245).

11. It seems that children from Entebbe were compared against the other 3 areas, but these were not compared among themselves. Please provide a brief explanation for this choice.

We chose to have Entebbe as the reference group because it was the largest group (and also the most urban location).

12. The LCA model is fitted separately at each time point: 1-5 years of age. This provides a lot of information, but can also originate some confusion: children born in Kigungu are less likely to be in LC 1 – and therefore have a higher probability for various infections – when they are 2 years of age, but the opposite occurs when they are 4 years of age (page 14, lines 242-247). This age dependency of LTA categorization needs to be discussed: is it a limitation of the study?

Thanks for raising this issue, we have now re-analysed all the 5 years’ data together, but also considered the first year analysis. We believe that this approach has simplified the results and eliminated the confusion.

13. SPT or IgE positivity is not equivalent of ARD. Please reformulate the definition of atopic and/or allergic outcome at 9y of age in the whole paper.

We thank the reviewers for this observation. We have now distinguished ARDs and atopy (SPT or asIgE) throughout the paper.

14. A table showing the prevalence of SPT/IgE positivity for each allergen extract should be given somewhere in the results. In particular, only a few extracts (blomia, mites and cockroach) are used in the presentation of the table 4. However, many more allergen extracts have been examined. Probably, the prevalence of e.g. pollen sensitization was low. However, this is not shown in the paper.

We thank the reviewers for this observation. Prevalence of SPT/IgE positivity for each allergen extract has been reported before in our earlier paper (Lule et al., 2017. Life-course of atopy and allergy-related disease events in tropical sub-Saharan Africa: A birth cohort study). We have briefly reported the same results again, in this manuscript (lines 326 – 333), and have included the prevalences of SPT positivity for peanut, Bermuda grass, cat, pollen and mould which we had initially skipped (lines 331 – 332). Indeed, prevalence of SPT positivity for each of these allergen extracts was less than 1.5%.